# Energy Harvester Based on an Eccentric Pendulum and Wiegand Wires

**DOI:** 10.3390/mi13040623

**Published:** 2022-04-15

**Authors:** Yi-Hsin Chen, Chien Lee, Yu-Jen Wang, You-Yu Chang, Yi-Cheng Chen

**Affiliations:** 1Department of Mechanical and Electro-Mechanical Engineering, National Sun Yat-sen University, Kaohsiung 804201, Taiwan; d093020004@nsysu.edu.tw (Y.-H.C.); m083020025@student.nsysu.edu.tw (Y.-Y.C.); 2Department of Intelligent Robotics, National Pingtung University, Pingtung 900392, Taiwan; clee@mail.nptu.edu.tw; 3Smart Sensing & Systems Technology Center, Industrial Technology Research Institute, Tainan 709410, Taiwan; benson_chen@itri.org.tw

**Keywords:** energy harvester, power generator, Wiegand wire, magnetic flux density

## Abstract

This study proposed an energy harvester that combines an eccentric pendulum with Wiegand wires to harvest the kinetic energy of a rotating plate. The energy harvester converts the kinetic energy into electrical energy to power sensors mounted on the rotating plate or wheel. The kinetic model is derived from the Euler–Lagrange equation. The eccentric pendulum generates a swing motion from the direction variation of the centrifugal force and the gravitational force. The magnetic circuit is designed such that, during the swing motion, an alternating magnetic field is formed to induce the output voltage of the Wiegand wire. COMSOL software was used to simulate magnetic flux density and optimize the geometric parameters of magnets. Response surface methodology was used to formulate the output voltage model. Magnetic flux density affects output voltage dramatically. However, the output voltage is not sensitive to the gradient of magnetic flux density. The experimental results indicate that when the Wiegand wire is 14.2 mm from the magnet, the generation power is 0.118–1.15 mW, in a speed range of 240–540 rpm. When the Wiegand wire is 7.0 mm from the magnet, the generation power is 0.741–1.06 mW, in a speed range of 480–660 rpm.

## 1. Introduction

To increase automobile driving safety, tire pressure monitoring systems (TPMSs) are used to monitor tire pressure in real time. At present, most TPMSs are powered by lithium batteries. However, due to their limited lifetime, lithium batteries are not environmentally friendly. With an increasing number of countries requiring that TPMSs be installed, interest in powering TPMSs from energy harvesters, rather than lithium batteries, is increasing [1,2]. Energy harvesters not only avoid potential environmental pollution, but also reduce maintenance costs. Energy harvesters that capture the energy of environmental vibrations to generate power include piezoelectric [3,4], electromagnetic [5,6], and electrostatic energy harvesters [7]. Energy harvesters applied to TPMSs extract energy from the changes in the components of the gravity force as the wheel rotates.

Zhiran et al. [8] proposed an energy harvester with an elastic piezoelectric bridge that can generate a maximum power output of 8.9 mW, a maximum output voltage of 7.3 V, and a maximum output current of 1.72 mA, under an optimal load of 3 kΩ. Wang et al. [9] proposed an energy harvester with a trapezoidal cantilever made of a piezoelectric material and connected a spring to the trapezoidal cantilever to automatically adjust the natural frequency of the system, to enhance the power generation and energy-harvesting efficiency. The design generated an average power of 99.78 μW, under an optimal load of 600 kΩ. Piezoelectric energy harvesting involves a piezoelectric material generating strain and producing electric charge; the output voltage is high, but the current is low. The efficiency of electromagnetic energy harvesting can be enhanced by modifying the magnetic circuit structure to have a high magnetic flux gradient to maximize output voltage. Wang [10,11] proposed an eccentric magnetic pendulum that uses the centrifugal force during the rotation of the wheel and the pendulum’s gravity to swing. The device generated a power of 200–442 μW, with rotations at 200–500 rpm. When an electromagnetic energy harvester is applied to an eccentric pendulum, the output voltage is usually lower than 0.7 V. This energy is difficult to store after diode rectification.

An electrostatic energy harvester can generate power by using two charged objects. The two charged objects exhibit a relative vibration that changes the capacitance and causes an electromotive force to generate power. Westby et al. [12] proposed a mini electret electrostatic energy harvester for use in a TPMS; at a car speed of 50 km/h, the harvester can produce 4.5 μW of output power. Yasuyuki Naito et al. [13] proposed a similar system that can produce up to 60 μW at a speed of 60 km/h. The weakness of this type of energy harvester is that it should be operated with a starting voltage and low-power generation efficiency.

This study chose to adopt the Wiegand effect as a potential TPMS power source. The Wiegand effect [14] is the instant switching of magnetization polarity inside a Wiegand wire, when the direction of the magnetic field outside the wire changes. The instantaneous magnetic field change on the Wiegand wire pickup coil when switching can generate high voltage pulses. The Wiegand wire must be driven with an alternating magnetic field (exchange in the N and S direction) and the alternating magnetic field lines must be in parallel with the Wiegand wire. The Wiegand wire is made of ${\mathrm{Fe}}_{0.4}{\mathrm{Co}}_{0.5}V_{0.1}$, with a diameter of 0.25 mm. After treatment, the outer coercivity is harder than the inner coercivity; the outer layer is generally called the “hard core”, and the inner layer is called the “soft core”. Takemura et al. [15,16,17,18] investigated the pulse characteristics of the output from a Wiegand wire, under different external magnetic field strengths. They found that the optimal operating scope of the Wiegand wire is within magnetic field strengths of 60–80 gauss. Chang et al. [19] also investigated the effects of different Wiegand wire magnetic flux densities. They revealed that the Wiegand wire is suitable for magnetic field strengths of 60–80 gauss. Maximum output voltage was obtained under external magnetic field strengths of 50–60 gauss, and output voltage decreased if the magnetic field was out of this range. Because of the high output voltage capabilities, an energy harvester mounted on a rotating object was developed using the Wiegand wires in this paper. In the following sections, a novel magnetic circuit was designed and analyzed for a pair of Wiegand wires at different distances to generate electrical energy. Complete governing equations were also derived.

## 2. Kinetic Equations of an Eccentric Pendulum

To harvest kinetic energy from a rotating wheel, this study proposed a design that installs a swing-type eccentric pendulum on the wheel rim or a plate, as illustrated in Figure 1. The components of gravitational force vary with respect to the pendulum as the wheel rotates.

To understand the dynamic behavior of the energy harvester, the kinetic equation of the energy harvester was derived from the Euler–Lagrange equation. The wheel center is set as the fixed coordinates (*x*_0_, 0), *R*_1_ is the wheel radius, and *R*_2_ is the distance from the wheel center to the rotational center of the eccentric pendulum. The swing angle of the eccentric pendulum is defined as *θ*, according to the moving coordinate ($\bar{x},\bar{y}$). The rotational angle of the wheel is defined as *Θ*, and the distance between the pivot joint and the mass center of the eccentric pendulum is defined as *r*. To describe the kinetic behavior of the energy-harvesting eccentric pendulum, with the origin situated at the rotational center of the eccentric pendulum, the coordinates are as follows:(1)$$\left\{\begin{array}{c} x_{1}=R_{2}cos\Theta \;\\ y_{1}=R_{2}sin\Theta \; \end{array}\right.$$
where the $\bar{x}$ axis is the extensive direction from the center of the rotating wheel to the pivot joint of the eccentric pendulum and the $\bar{y}$ axis is the tangential direction of the wheel rim. Therefore, the coordinates of the mass center of the energy-harvesting eccentric pendulum are:(2)$$\left\{\begin{array}{c} x_{2}=R_{2}cos\Theta +rcos\left(\Theta +\theta \right)\\ y_{2}=R_{2}sin\Theta +rsin\left(\Theta +\theta \right) \end{array}\right.$$

The kinetic energy of the eccentric pendulum is:(3)$$T_{h}=\frac{1}{2}m\left({\dot{x_{2}}}^{2}+{\dot{y_{2}}}^{2}\right)+\frac{1}{2}I_{h}{\left(\dot{\theta }+\dot{\Theta }\right)}^{2}$$
where *m* and $I_{h}$ are the mass and mass moment of inertia of the eccentric pendulum, respectively. The kinetic energy of the rotational wheel is:(4)$$T_{w}=\frac{1}{2}I_{w}{\dot{\Theta }}^{2}$$
where $I_{w}$ is the moment of inertia of the wheel. The gravitational potential energy of the eccentric pendulum is:(5)$$V=mgy_{2}$$
where *g* is the acceleration of gravity. The damping-dissipated energy of the energy harvester is defined as:(6)$$D_{T}=\frac{1}{2}c_{T}{\dot{\theta }}^{2}$$
where *c_T_* is the damping constant of the eccentric pendulum. The Lagrangian mechanics of the entire system *L* are defined as:(7)$$L=T_{h}+T_{w}-V$$

From the Euler–Lagrange equation:(8)$$\frac{d}{dt}\left(\frac{dL}{d\dot{\theta }}\right)-\frac{dL}{d\theta }+\frac{dD_{T}}{d\dot{\theta }}=0$$

The kinetic equation of the eccentric pendulum is:(9)$$\left(I_{h}+mr^{2}\right)\ddot{\theta }+\left(\left(I_{h}+mr^{2}+mrR_{2}\cos\theta \right)\ddot{\Theta }+gmrcos\left(\theta +\Theta \right)+mrR_{2}{\dot{\Theta }}^{2}\sin\theta \right)=-c_{T}\dot{\theta \;}$$

If the vehicle is at a constant speed, $\ddot{\Theta }=0$ is taken and the eccentric pendulum is assumed to be swinging within a relatively small angle, sinθ≒θ and cosθ≒1. Therefore, Equation (9) is simplified as follows:(10)$$\ddot{\theta }+\frac{1}{I_{h}+mr^{2}}\left(gmrcos\left(\theta +\Theta \right)+mrR_{2}{\dot{\Theta }}^{2}\theta \right)=-c_{T}{}^{*}\dot{\theta }$$
where the natural frequency of the eccentric pendulum is derived as follows:(11)$$\omega_{n}=\dot{\Theta }\sqrt{\frac{R_{2}r}{r^{2}+k_{\lambda }^{2}}\;}$$
where *k_λ_^2^ = I_h_/m*. Here, the characteristic length of the system *L** is defined as follows:(12)$$L^{*}=\frac{r^{2}+k_{\lambda }^{2}}{r}$$

Then, Equation (11) is computed as follows:(13)$$\omega_{n}=\dot{\Theta }\sqrt{\frac{R_{2}}{L^{*}}}\;$$

The foregoing computes the natural frequency of the eccentric pendulum. With a proper design to satisfy *L** = $R_{2}$, the natural frequency of the eccentric pendulum equals the angular frequency of the wheel at any car speed. That is, resonance can be approached at any wheel rotation speed to obtain highly efficient energy harvesting.

### Power Generation Analysis of the Eccentric Pendulum

To evaluate the response of the energy harvester under different dynamic parameters, the kinetic equation of the energy harvester was solved using numerical methods. According to the law of energy conservation, the electrical power gained by the energy harvester is identical to the power dissipated by the damper. For *L** = 0.203 m, the average electrical power at different wheel speeds varies with $c_{T}^{*}=\frac{C_{T}}{I_{h}+mr^{2}}$, as illustrated in Figure 2. Increasing wheel rotational speed increases electrical power; each rotation speed corresponds to an optimal damper value to attain the maximum electrical power. Increasing rotation speeds reduce the optimal damper values; the optimal damper value range under rotations of 200–600 rpm is 1.5–2.5 N-s/kg/m. For example, $c_{T}^{*}$ = 1 N-s/kg/m in ideal conditions produces 1.82–3.89 mW.

## 3. Wiegand Wire in a Magnetic Field

WG631 Wiegand wires made by Nanjing AH Electronic Science & Technology were used (Figure 3). The amplitude of the Wiegand wire output voltage is not related to the velocities of the magnetic field change and needs to be aligned with the north–south (NS) poles of alternating magnetic fields (AMFs) to induce voltage.

The Wiegand wire is excited by AMFs with the nonlinear hysteresis effect. This study proposed a magnetic circuit design, as illustrated in Figure 4. The Wiegand wire in the center is at the initial position and is combined with two magnets that contribute the same magnetic field strengths but in the opposite direction. The arrangement ensures that the magnetic flux of the Wiegand wire is zero at the initial position. Moving left and right applies AMFs of N and S to the Wiegand wire, respectively. This magnetic circuit design was used for the eccentric pendulum; specifically, magnets are mounted on the eccentric pendulum and the Wiegand wire is placed underneath. As the eccentric pendulum swings, AMFs excite the Wiegand wire to produce voltage.

The magnetic field lines should be parallel to the Wiegand wire. Therefore, the magnetic field lines start from the N pole and follow along the wire’s axial direction back to the S pole to complete a loop simulated by COMSOL, as illustrated in Figure 5. This indicates that the magnetization direction of the Wiegand wire can be altered by the small displacement of the magnet pair. In the simulation, physics-controlled mesh and a prism element were selected. The resulting mesh consists of about 19,716 elements. The residual induction of the NdFeB magnets was 1.4 T, and the relative permeability of the Wiegand wire was 20. The B–H curve of the Wiegand wire, based on experimental determination, is shown in Figure 6.

The amplitudes of the voltage output from the Wiegand wire are irrelevant to the shifting magnetic field velocities. The linear stage is used, as illustrated in Figure 7, and the geometric definitions of the magnet arrangements are listed in Figure 7B. *D* is the distance that the paired magnets move back and forth. For example, given *D* = 1 mm, the paired magnets move 1 mm leftwards and rightwards from the Wiegand wire (the origin). For *D* = 1 mm, the output voltages of the Wiegand wire for two distance parameters (*b*, *d*), under different linear stage velocities (for velocities of different magnetic field changes), are illustrated in Figure 8. Within a linear stage moving velocity of 2–20 mm/s, the output voltages were measured to be identical for each magnet arrangement, which indicates that the shifting magnetic field velocities are irrelevant to the output voltages. As illustrated in Figure 8, the parameters *b* and *d* affect the strength of the magnetic field to change the output voltage.

### 3.1. Output Voltage Varied with Magnetic Flux Density

To derive a power generation model, experiments for different magnetic field strengths and gradients were needed to determine the Wiegand wire output voltage. For *b* = 12 mm, *d* = 4.3–8.3 mm, and *D* = 1–10 mm, the experimental output voltages are listed in Figure 9. Different magnetic circuit parameters (*b*, *d*, *D*) represent the differences in magnetic flux densities applied to the Wiegand wire. The results of using COMSOL finite element software to simulate the axial average magnetic flux density *B_w_* in the Wiegand wire for different *b*, *d*, and *D* are presented in Figure 10, and the spatial gradient of the axial average magnetic flux density *B_w_/D* is illustrated in Figure 11. Using Figure 10 and Figure 11, the correlations between the magnetic flux densities and the output voltages in the experiments were obtained, as illustrated in Figure 12. Given *d* = 4.3–8.3 mm and a maximum output voltage of 1.44 V, the *B_w_* value was 0.12 T. For any value higher or lower than *B_w_* = 0.12 T, the output voltage quantitatively declines. The gradient of the magnetic flux density (*B_w_/D*) weakly affected the output voltage. Within 0.8 times the maximum output voltage amplitude, an output voltage range of WG631 of 1.12–1.44 V corresponded to a magnetic flux density range of 0.06–0.22 T, as illustrated in Figure 12.

### 3.2. Response Surface Methodology for V, B_w_, and B_w_/D

Response surface methodology (RSM) was used to obtain the nonlinear relationship of the output voltage *V* of the Wiegand wire, the magnetic flux densities of the Wiegand wire *B_w_*, and the gradients of the magnetic flux densities of the Wiegand wire *B_w_/D* for subsequent simulations and predictions of the output voltage. RSM was used as the purpose for getting the approximation function of the independent parameter and output. Figure 13 revealed that the *B_w_* variable had an optimal value that corresponded to the *V* maximum. No overfitting was found in the zones without the data points of the model.

### 3.3. Sensitivity Analysis

Sensitivity analysis was conducted based on the differentiating RSM model, with respect to the magnetic flux density, as shown in Figure 14. The high *B_w_-V* sensitivity ($\frac{\partial V}{\partial B_{W}}$*)* corresponding to a low *B_w_* value was within a sensitivity range of −20 to 40 V/T. The mean value of *B_w_-V* was 8.51 V/T. For comparison, the *B_w_/D-V* sensitivity was calculated by differentiating the RSM model corresponding to the magnetic flux density gradient, as shown in Figure 15. The *B_w_/D-V* sensitivity value was in the range of −16 to 10 V/(T/mm) for the same ranges of *B_w_/D* and *B_w_*.

### 3.4. Optimization of the Magnetic Field Design

A COMSOL optimization module was used to identify the optimized geometric parameters of the magnets applied to the energy harvester. Bound optimization by quadratic approximation is an iterative algorithm for finding a minimum of an objective function, subject to bounds on the variables, and was selected as the optimization method in this study. The eccentric pendulum device consisted of one pair of fan magnets to induce the Wiegand wire through AMFs as the paired magnets swing. The geometric parameters are displayed in Figure 16. To minimize the geometric parameter numbers in the optimization process, the ratio was defined as *t_s_/w* and a constant volume was considered. The swing angle was set to 3°. Given a fixed weight, the objective function was chosen to maximize the axial magnetic flux densities *B_wmax_* induced by AMFs in the Wiegand wire for fan magnets swinging 3°. The outer radius of fan *r_o_* was fixed at 18 mm, the magnet area *A_s_* was fixed at 50 mm^2^ and $d_{s}$ was fixed at 7 mm. The parameters that were to be optimized were the gap *b*_s_, the width *t_s_*, and *t_s_/w*, in which *w* depends on *t_s_*. The upper limits of *b*_s_ and *t_s_* were 10 mm each.

In the optimization process, a global maximum could depend on initial values due to different search paths, as a result of the gradient of the global maximum being too small. Table 1 presents the optimization results using different initial values. The maximum *B_w_* within a range of 0.13489–0.13491 T corresponded to different *t_s_* and *b_s_* values. The results are attributed to a small gradient in the global maximum. Figure 17 presents how the contour map of *B_w_* varied with *t_s_/w* and *b_s_*. The large yellow plateau region reflects the flat gradient of the global maximum. The parameters of *b*_s_= 2.25 mm, *t_s_* = 4.73 mm, and *t_s_/w* = 0.389 were selected as the optimized design.

### 3.5. Cogging Torque

Cogging torque (noncurrent torque) is induced by magnets approaching and deviating from permeable materials. In terms of simulations of the fan magnets swinging using COMSOL, the cogging torque figures produced by the Wiegand wire are shown in Figure 18, with cogging torque as *T_cog_ = f*(*θ*) by using a sin function. Cogging torque relates to the rotation angle of the fan magnets. The cogging torque value is 0 under a zero-degree magnet swing angle due to equal contributions from both magnetic fields. The cogging torque *T_cog_* regressed by the polynomial equation was added in the kinetic equation of the eccentric pendulum Equation (9).

## 4. Results

A prototype of the energy harvester, according to the optimized design parameters in Section 3.4, was fabricated, as displayed in Figure 19. The NdFeB fan magnets were fabricated by Supermax Co., Ltd (Kaohsiung, Taiwan, China). The residual induction was 1.4 T. The experimental setup is presented in Figure 20. The energy harvester was mounted on the rotary plate. For a constant rotation speed, the kinetic model of rotational motion equals that of rolling motion. The output voltage produced by the Wiegand wire was transmitted by a slip ring and sensed by a NI USB-9234 data acquisition card. The eccentric pendulum produced different swing amplitudes, caused by different rotation speeds, indicating that the Wiegand wire experienced different magnetic flux densities. The generalized damping constant *C_T_** includes electrical and mechanical dampers that are attributed to frictional torque from bearings and the air. The mechanical damper was estimated by the swing angles in the open-circuit Wiegand wire. The mechanical damper was approximately 1.5 N-s/kg/m. The output voltage responses of the Wiegand wire are displayed in Figure 21. Based on the impedance matching, the resistance value of the external load used in the experiment was 340 Ω, which is the same as the resistance value of the WG631 Wiegand wire.

Two Wiegand wires with different *ds* were embedded in the energy harvester because of different swing angles at low and high rotation speeds. The eccentric pendulum produces large swing angles due to small centrifugal forces, caused by low rotation speeds. Therefore, a large distance between the Wiegand wire and magnets *d_s_* = 14.2 mm was set so that power would be generated at low rotation speeds. The swing angle at low rotation speeds approached 3°, as calculated in the optimization process. The distance *d_s_* = 7.0 mm considered in the optimization process was chosen for high rotation speeds. By combining the kinetic equation, magnetic circuit simulation, RSM model, and the cogging torque, this study obtained the simulated output voltage, which is displayed in Figure 22. For *d_s_* = 7.0 mm, the experimental results indicate that the Wiegand wire operated at the optimal operating point and produced an output voltage larger than 0.7 V within rotations of 480–660 rpm. The prototype using *d_s_* = 14.2 mm and *d_s_* = 7.0 mm on two Wiegand wires simultaneously achieved an output voltage larger than 0.7 V within a wider rotation scope (240–660 rpm). This output voltage is large enough to be used for bridge rectifier circuits to exceed the diode forward voltage. Once the voltage generated by an energy harvester is larger than 0.7 V, the bridge rectifier cannot provide output voltage for the storage circuit.

The power computational results are illustrated in Figure 22. Given *d_s_* = 14.2 mm and rotations of 240–540 rpm, the power generation range was 0.118–1.15 mW. Given *d_s_* = 7.0 mm and rotations of 480–660 rpm, the power generation range was 0.741–1.06 mW. In Figure 23, the output powers of both Wiegand wires are superimposed and shown as the black line. The output power was enhanced for wider rotation ranges. The differences between experiments and simulations are mainly attributed to the nonlinear frictional torque from the ball bearing mounted on the eccentric pendulum. The nonlinear frictional torque was accompanied by the centrifugal force in the rotational motion. The nonlinear frictional torque in the energy harvester will be explored in the future.

## 5. Conclusions

This study proposed an energy harvester using Wiegand wires for a rotating object. The kinetic equations of a round eccentric pendulum mounted on a rotating object were derived via the Euler–Lagrange equation. Computational results of the natural frequency of the energy harvester were obtained, and the characteristic length *L** was defined to show the effects on the swing behaviors of the eccentric pendulum. In the simulation results, when *L* = R_2_*, the natural frequency of the energy harvester was identical to the rotation frequency of the rotary plate, which makes the eccentric pendulum approach resonance status. The magnetic circuit design used magnets to convert kinetic energy exerted by the swinging eccentric pendulum to electrical energy through the Wiegand wire moving in the magnetic field. An experimentally determined model for the magnetic flux densities, the magnetic flux density gradients, and the output voltage of the Wiegand wire was provided by using the RSM.

In the experiment where a rotating plate was used to generate voltage, some disparities between the model-based predictions and the experimental results were observed. The Wiegand wire generated a mean impulse voltage of 1.19 V at 240–660 rpm. Discrepancies between the experimental and simulated results were attributed to unpredictable and nonlinear frictional torque. When the rotation speed was higher than 800 rpm, the swing angle decreased dramatically due to large frictional torque. If the magnet swings at small and asymmetrical angles, the half period cannot generate voltage. Because of different swing angles at different rotation speeds that induce large magnetic flux density variation, using two Wiegand wires, one close to and one far from the magnets, extends the workable speed range. Using two Wiegand wires connected in series increased electrical power generation to 0.88–1.24 mW, within rotations of 240–660 rpm. The experimental results indicate that the proposed energy harvester could be a potential power source for TPMS.

## Figures and Tables

**Figure 1 micromachines-13-00623-f001:**
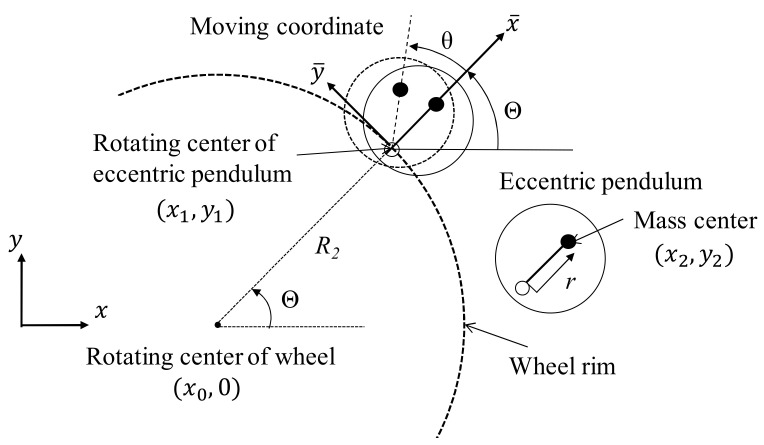
Eccentric pendulum installed on a wheel rim.

**Figure 2 micromachines-13-00623-f002:**
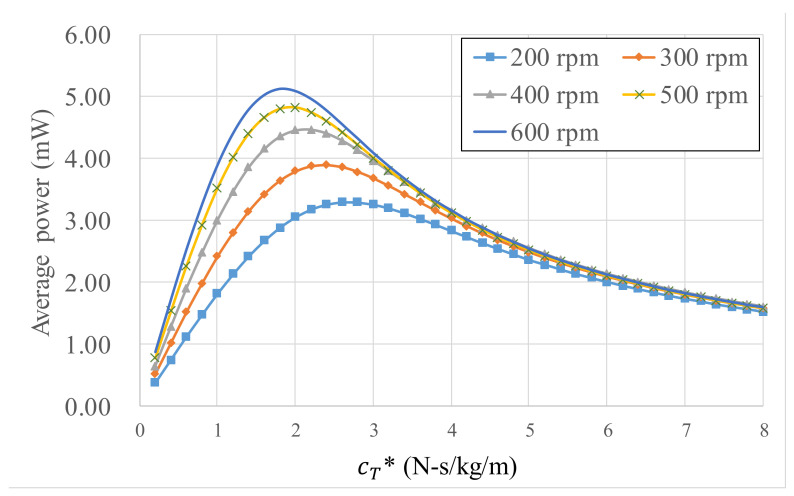
Electrical power varies with *C_T_** at different wheel rotation speeds.

**Figure 3 micromachines-13-00623-f003:**
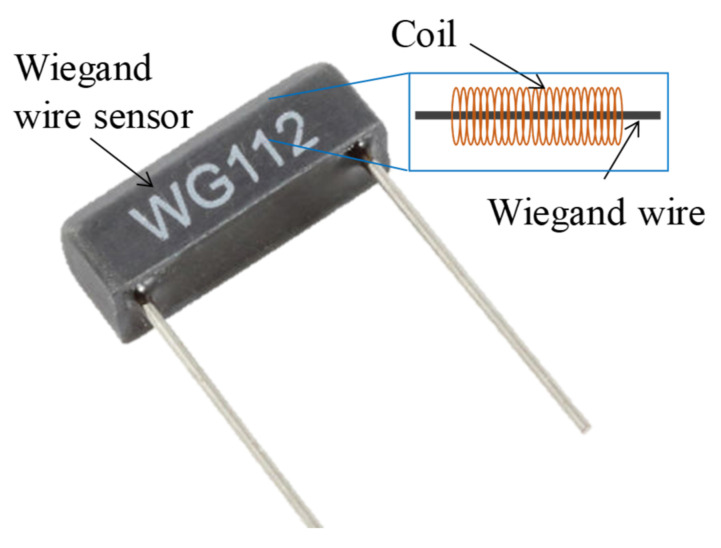
WG631 Wiegand wire.

**Figure 4 micromachines-13-00623-f004:**
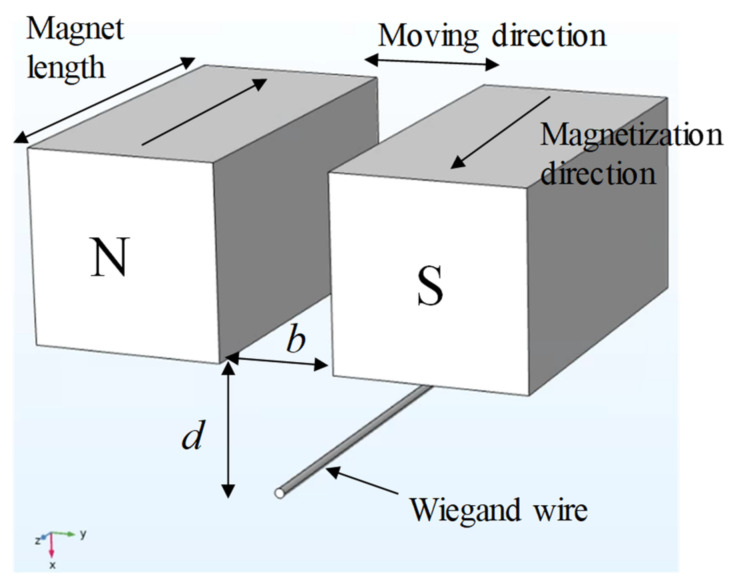
Magnetic circuit design and definitions.

**Figure 5 micromachines-13-00623-f005:**
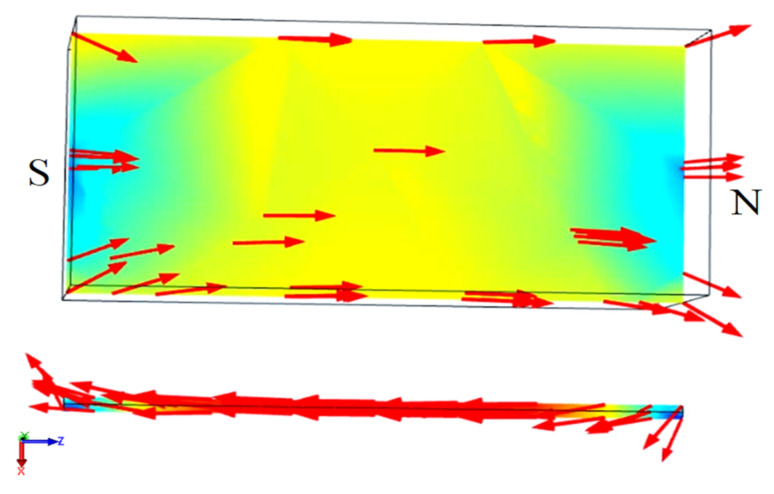
Magnetic field line simulation for a magnet parallel to a Wiegand wire.

**Figure 6 micromachines-13-00623-f006:**
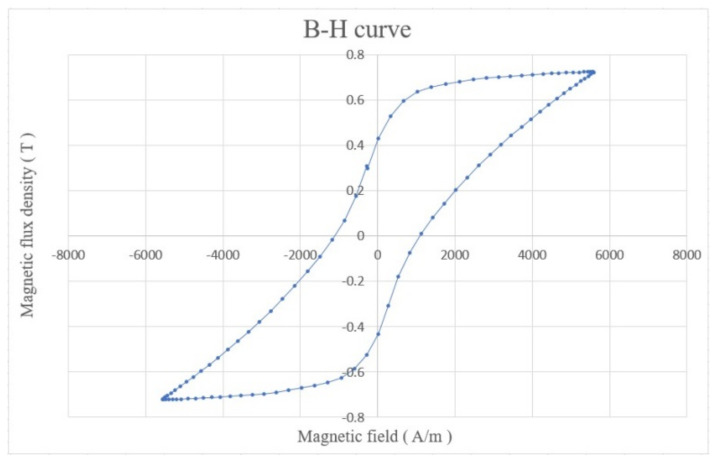
B–H curve of the Weigand wire.

**Figure 7 micromachines-13-00623-f007:**
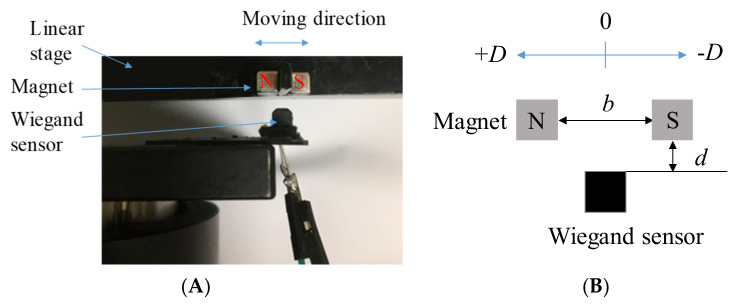
Using a linear stage to test Wiegand wires. (**A**) Experimental setup. (**B**) Parameter definitions.

**Figure 8 micromachines-13-00623-f008:**
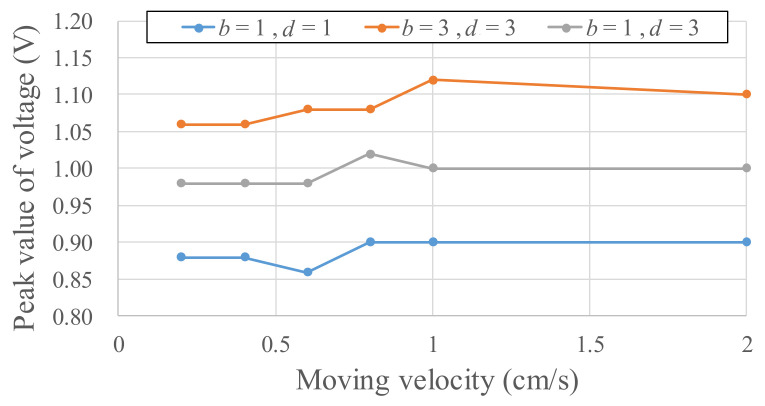
Output voltages of the Wiegand wire under different magnetic field parameters.

**Figure 9 micromachines-13-00623-f009:**
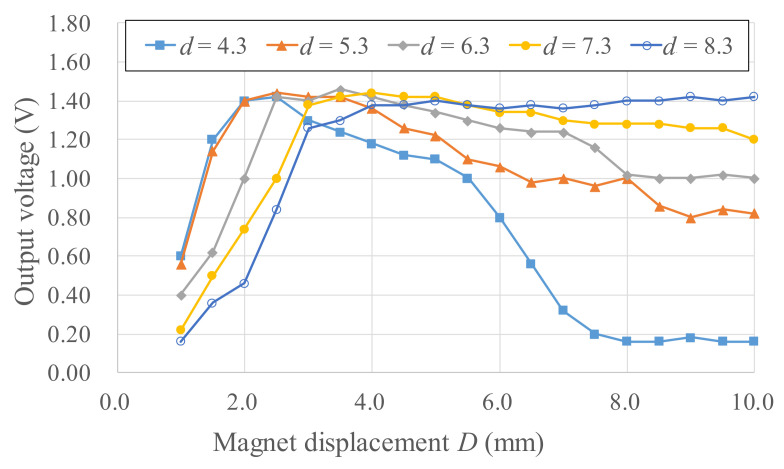
Output voltages of the Wiegand wire under different magnetic field strengths and gradients.

**Figure 10 micromachines-13-00623-f010:**
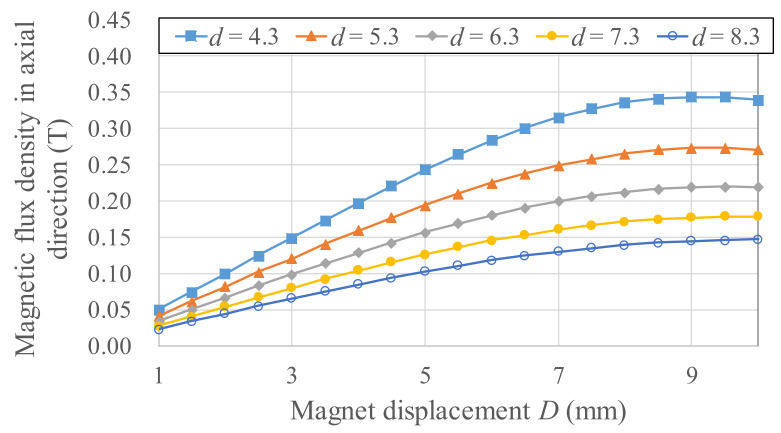
Results of using COMSOL Multiphysics to simulate *b*, *d*, and *D*, the Wiegand wire axial average magnetic flux density.

**Figure 11 micromachines-13-00623-f011:**
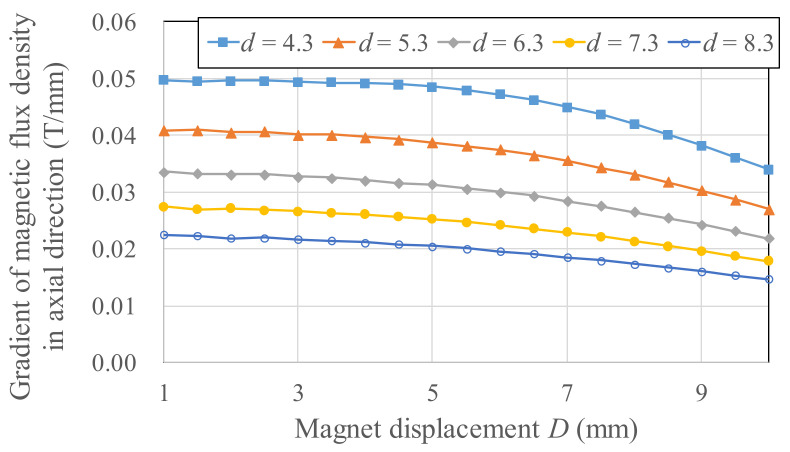
Spatial gradient of the axial average magnetic flux density *B_w_/D* for the Wiegand wire.

**Figure 12 micromachines-13-00623-f012:**
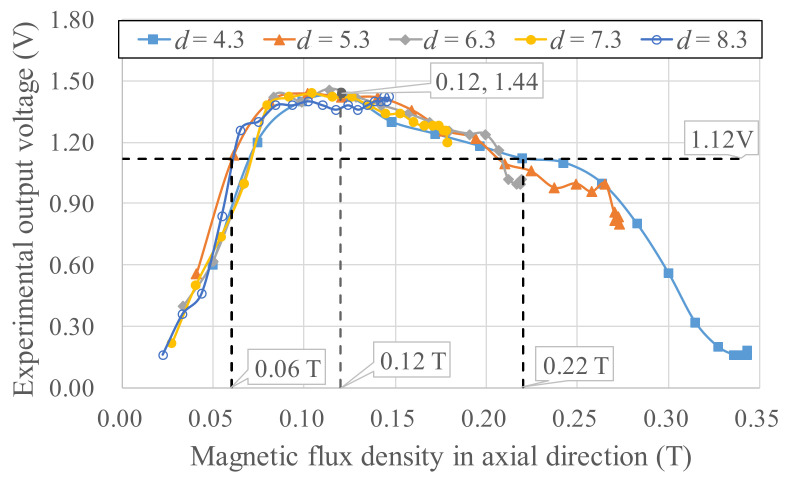
Correlations between the magnetic flux densities and the output voltages in the experiments.

**Figure 13 micromachines-13-00623-f013:**
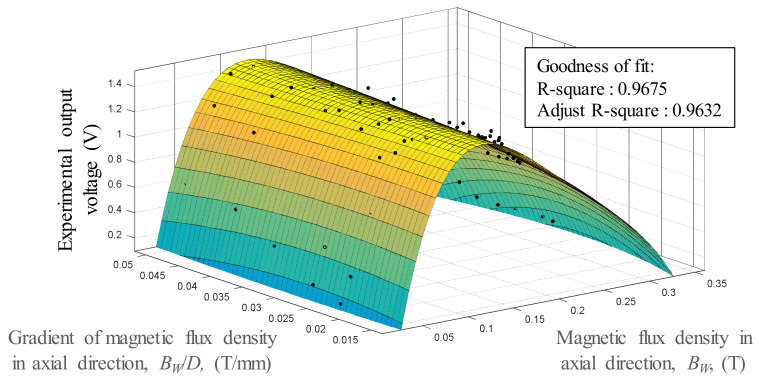
Response surface methodology model expresses the relationships between *V*, *B_w_*, and *B_w_/D*.

**Figure 14 micromachines-13-00623-f014:**
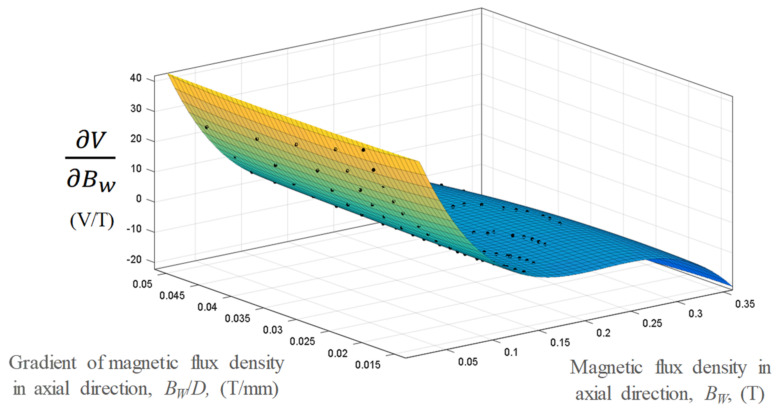
Sensitivity analysis of the variation in *B_w_-V* with *B_w_/D* and *B_w_*.

**Figure 15 micromachines-13-00623-f015:**
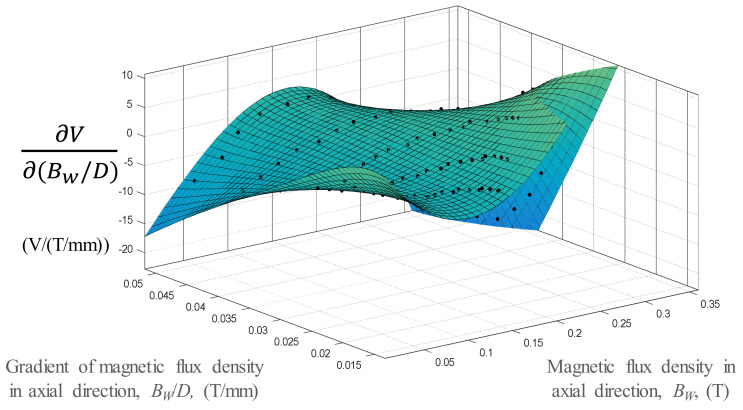
Sensitivity analysis of the variation in *B_w_/D-V* with *B_w_/D* and *B_w_*.

**Figure 16 micromachines-13-00623-f016:**
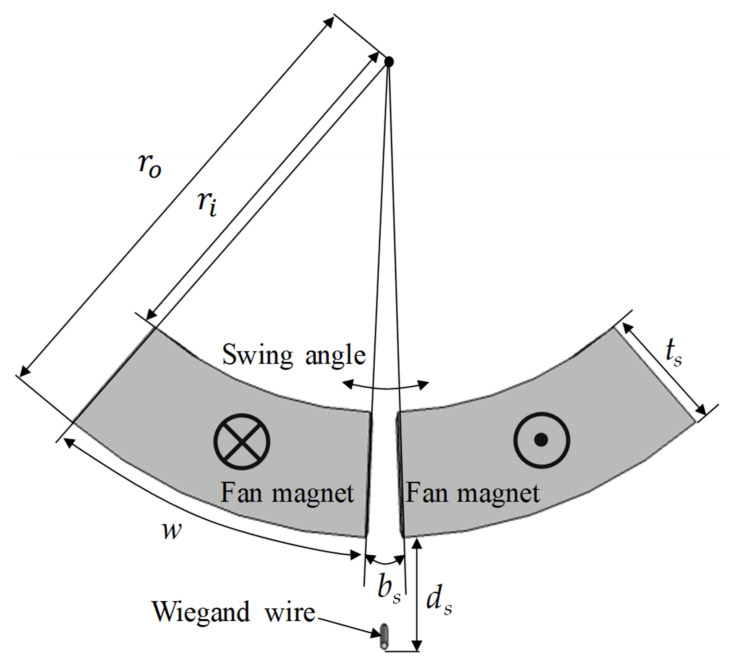
Geometric parameters and polarization directions of the magnet pair for the optimization process.

**Figure 17 micromachines-13-00623-f017:**
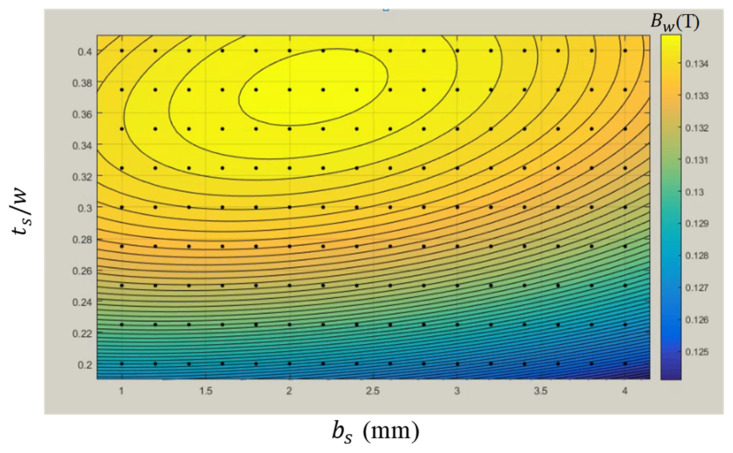
Contour map of *B_w_* varied with *b_s_* = 1–4 mm and *t_s_/w* = 0.2–0.4.

**Figure 18 micromachines-13-00623-f018:**
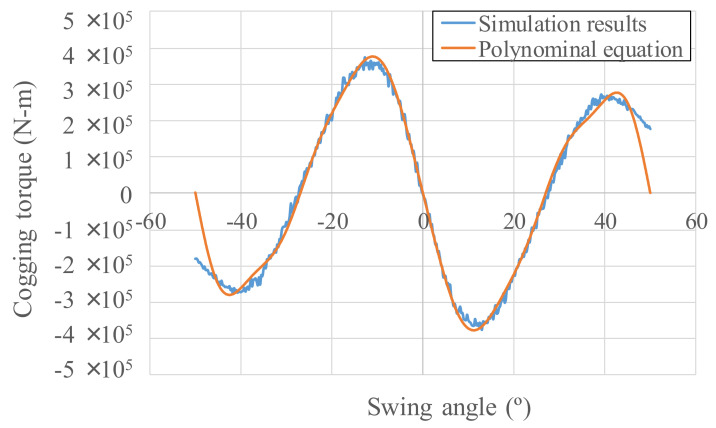
Cogging torque simulation and regression results.

**Figure 19 micromachines-13-00623-f019:**
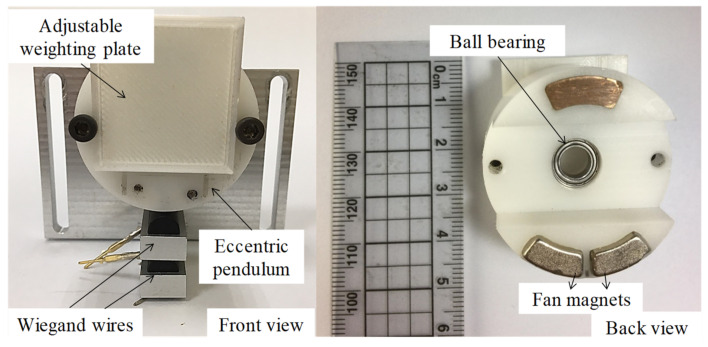
Energy harvester composed of fan magnets and Wiegand wires.

**Figure 20 micromachines-13-00623-f020:**
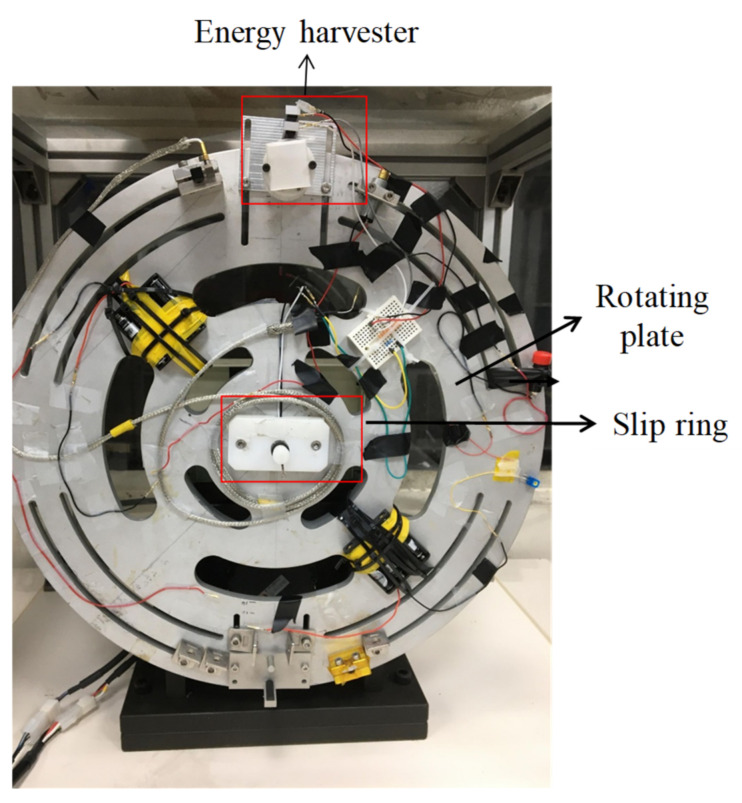
Experimental setup for energy harvesting.

**Figure 21 micromachines-13-00623-f021:**
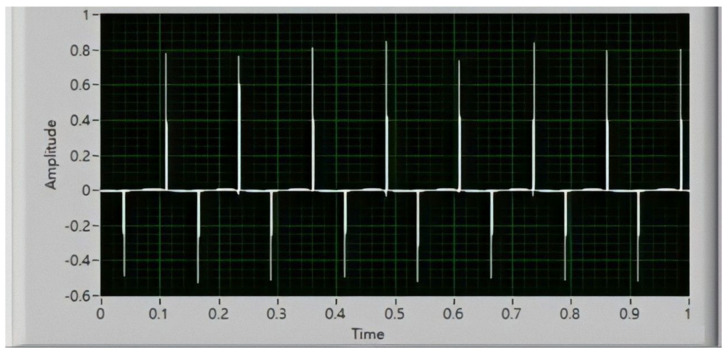
Experimental output voltage of the Wiegand wire.

**Figure 22 micromachines-13-00623-f022:**
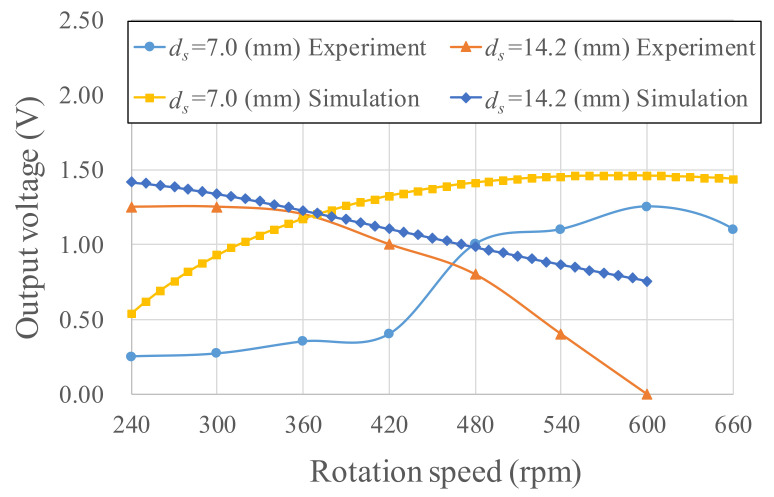
Simulated and experimental voltage of the proposed energy harvester.

**Figure 23 micromachines-13-00623-f023:**
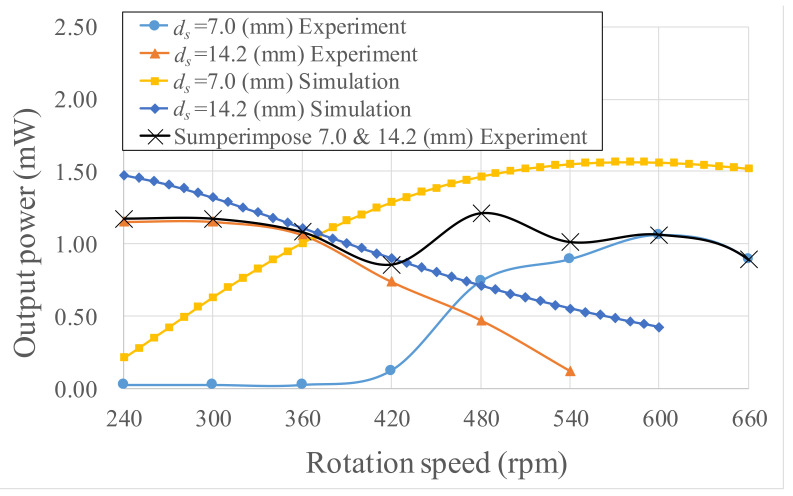
Simulated and experimental power of the proposed energy harvester.

**Table 1 micromachines-13-00623-t001:** Optimization Results of Multiple Initial Values.

	Optimization Result			Other Result	
Initial Values (mm)	$t_{s}\;(\mathrm{mm})$	$b_{s}\;(\mathrm{mm})$	$B_{wmax}\;(T)$	w (°)	$t_{s}/w$
$b_{s}$$=3,\;t_{s}$ = 3	4.58	2.12	0.13489	40.24	0.362
$b_{s}$$=5,\;t_{s}$ = 5	4.73	2.25	0.13491	38.75	0.389
$b_{s}$$=7,\;t_{s}$ = 7	4.69	2.32	0.13489	39.13	0.382
$b_{s}$$=9,\;t_{s}$ = 9	4.60	2.04	0.13491	40.00	0.366

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
