# Peer review of "Energy Harvester Based on an Eccentric Pendulum and Wiegand Wires"

_micromachines, 2022, doi:10.3390/mi13040623_

Round 1

Reviewer 1 Report

[1] Did you use ANSYS or COMSOL or both? I noticed that you used ANSYS in line 201. But others used COMSOL. Would you please put COMSOL and ANSYS in references? If it is possible, please provide ansys and comsol code for the supporting materials.

[2] It will be clearer if you disclose the materials properties and input parameters for comsol/ansys in table. Also provide type of element, meshing type, number of elements, etc.

[3] Would you provide the magnetic hysteresis curve of Wiegand wire? You can put that on the supporting document.

[4] Would you please provide the magnet material name ( also included company name or trademark), disclosed magnetic properties, geometric parameters in tables?

[5] Please add the annotations to clarify your Figure 15. Not only the geometric annotations.

[5] Fig. 21 and Fig. 22 show that the experiments and simulated result are very different. What are the reasons? Can you improve the simulated result or experimental results?

Author Response

We are most grateful for the reviewers’ comments, which have helped us to improve our manuscript. We have also responded to the reviewers’ comments point-by-point below and highlighted the revised and added parts in red.

Reviewer 2 Report

This paper proposed an energy harvester that combines an eccentric pendulum with Wiegand wires to harvest the kinetic energy of a rotating plate. And the output characteristics are studied based on the established kinetic model using COMSOL and ANSYS. However, the following details need to be improved:

1.Pay attention to the use of parameters: there is ‘g’ in equation (5), but it is used by a different form when explained at line 126; the same problem exists in equation (11) and line 153; and ‘ds’ at line 337.

2.‘CT’ was used to express two meanings in Equations (6) and (9), one is ‘the electric-load damping of the energy harvester’(line 130), and the other is ‘the damping constant of the eccentric pendulum’(line 143).

3.Units and diagrams: Check the units between Line 174 and Figure 2; and unit of ordinate in Figure 17. And there are three representations of figures in this paper, for example, ‘Figure 4’, ‘Figure.13’ and ‘Fig.14’.

4.Check figure 11 and ensure the integrity of drawing parameters ‘d’. And from the line 235-240, ‘a maximum output voltage of 1.4V, the Bw value was 0.12T’, but from the figure 11, when the Bw=0.12T, the voltage is 1.44V.

5. The figure quality in the paper needs to be improved.

Author Response

(The authors gave the same response as above.)

Round 2

Reviewer 1 Report

Thank you for improving the quality of your own mansucript.
I have my final comments as the following comments:
[1] Figure 16: should add the annotations related to the polars of fan magnets.
[2] Figure 21. would you please use the Labview data and plot it for a nicer and clearer plot?
[3] Line 405 - 410: please try to improve the simmulation with your assumptions and take time to improve the quality of your own manuscript for your own career.
[4] I did not see any supporting documents in the review system. Please provide them.

I hope you will improve the quality of your manuscript for yourselves. 
I leave my decision for the authors after your final revision. 
If you did do your best, it could be published.

Best regards,
LVM

Reviewer 2 Report

In my opinion, this article can be accepted for publication.

Author Response

Dear Reviewer,

We all appreciate your effort to review the paper and to enhance the quality.